# Stability of Chimpanzee Adenovirus Vectored Vaccines (ChAdOx1 and ChAdOx2) in Liquid and Lyophilised Formulations

**DOI:** 10.3390/vaccines9111249

**Published:** 2021-10-28

**Authors:** Adam Berg, Daniel Wright, Pawan Dulal, Anna Stedman, Sofiya Fedosyuk, Michael J. Francis, Bryan Charleston, George M. Warimwe, Alexander D. Douglas

**Affiliations:** 1Wellcome Trust Centre for Human Genetics, Jenner Institute, University of Oxford, Roosevelt Drive, Oxford OX3 7BN, UK; adam.berg@ndm.ox.ac.uk (A.B.); pawan.dulal@ndm.ox.ac.uk (P.D.); sofiya.fedosyuk@ndm.ox.ac.uk (S.F.); 2The Pirbright Institute, Ash Road, Pirbright, Woking GU24 0NF, UK; anasteadman@hotmail.com (A.S.); bryan.charleston@pirbright.ac.uk (B.C.); GWarimwe@kemri-wellcome.org (G.M.W.); 3BioVacc Consulting Ltd., The Red House, 10 Market Square, Amersham HP7 0DQ, UK; mike.francis@biovacc.com; 4KEMRI-Wellcome Trust Research Programme, Kilifi P.O. Box 230-80108, Kenya; 5Centre for Tropical Medicine & Global Health, University of Oxford, Oxford OX3 7LG, UK

**Keywords:** stability, vaccine formulation, adenovirus

## Abstract

Adenovirus vectored vaccines have entered global use during the COVID-19 pandemic, and are in development for multiple other human and veterinary applications. An attraction of the technology is the suitability of the vaccines for storage at 2–8 °C for months. Widely used COVID-19 vaccine ChAdOx1 nCoV-19 (University of Oxford/AstraZeneca) is based on a species E simian adenovirus. Species E simian serotypes have been used in a wide range of other development programs, but the stability of such vectors has not been extensively described in the peer-reviewed literature. Here, we explore the stability of two candidate vaccines based on two species E serotypes: a Rift Valley fever vaccine based upon the ChAdOx1 vector (Y25 serotype) used in ChAdOx1 nCoV-19, and a rabies vaccine based upon a ChAdOx2 vector (AdC68 serotype). We describe each vector’s stability in liquid and lyophilised formulations using in vitro and in vivo potency measurements. Our data support the suitability of liquid formulations of these vectors for storage at 2–8 °C for up to 1 year, and potentially for nonrefrigerated storage for a brief period during last-leg distribution (perhaps 1–3 days at 20 °C—the precise definition of acceptable last-leg storage conditions would require further product-specific data). Depending upon the level of inprocess potency loss that is economically acceptable, and the level of instorage loss that is compatible with maintenance of acceptable end-of-storage potency, a previously reported lyophilised formulation may enable longer term storage at 20 °C or storage for a number of days at 30 °C.

## 1. Introduction

Adenovirus vectors offer a versatile vaccine platform that is now in widespread use for COVID-19, licensed for Ebola, and in development for a range of other indications [1,2,3]. Suitability for distribution and storage at 2–8 °C for months is an advantage of the platform as compared to some alternative vaccine technologies, and is significantly important in reaching underserved communities and people in countries with poor capacity for the distribution of frozen products.

There is a fairly extensive peer-reviewed and patent literature relating to the storage of adenoviruses in a wide variety of formulations, mostly relating to human adenovirus serotypes from species B, C, and D [4]. For ICH-compliant product stability studies, in vitro infectivity assays are frequently used, but in vivo potency measurement, as reflected by immunogenicity in mice, is considered in some quarters to be a gold standard.

A family of liquid formulations developed by Evans, Volkin, and colleagues (Merck & Co., Inc., Kenilworth, NJ, USA) are among the most widely used in the field: Evans’ A195 formulation is sometimes regarded as a benchmark, in particular for human adenovirus serotype 5 (AdHu5; species C) [5,6]. Common ingredients of these formulations are sucrose, polysorbate-80, magnesium chloride, a pH buffer (either Tris or histidine), ethanol, and EDTA. Formulations reported by other authors also share some of these excipients, in particular sucrose and polysorbate-80 [7,8]. Generally, such buffers permit long-term storage at 2–8 °C. Some authors reported that other additives, including amino acids and polyethylene glycol, may enhance stability at 20–25 °C, though these studies used relatively low concentrations of the virus, which may have a favourable effect on stability by reducing aggregation [9,10]. To our knowledge, no buffer has reproducibly demonstrated potential to support long-term storage of virus at concentrations relevant to clinical dosing (c. 1 × 10^11^ VP/mL) at temperatures above 2–8 °C.

Lyophilised formulations have been used for many years for human and veterinary live-attenuated adenoviral vaccines [11,12], but do not typically overcome the need for storage at 2–8 °C. For adenovirus vectored vaccines, lyophilisation has not yet found widespread use. The peer-reviewed literature discloses formulations and cycle parameters that achieve stability at temperatures up to 37 °C for periods of around one month [13,14,15]. Further data are reported in the patent literature and in publications that do not fully disclose the details of the excipient composition [7,16].

An advantage of the adenoviral platform is the generally similar stability performance of vaccines based upon a single serotype regardless of the delivered antigen, and hence the predictability of the stability of a novel vaccine (critical in outbreak response). This is because the antigen is not a structural protein in the virion (merely encoded by the genome) and appears to hold true unless the size of the packaged genome deviates excessively from that of the wild-type adenovirus [17]. More broadly, because adenoviral serotypes within a species group have relatively similar capsid proteins, their stability performance in a given buffer is likely to be more similar to each other than to vectors from a different species group.

In contrast to the extensive literature on the stability of human adenoviruses, there are relatively few published data on the stability of species E adenoviruses. Species E include many of the simian adenoviruses that have been used as vaccine vectors, notably Y25 (the basis of the University of Oxford’s ChAdOx1 platform) [18], AdC68 (also known as SAdV-25, and the basis of the University of Oxford’s ChAdOx2 platform) [19], and ChAd63 [20]. ChAdOx1 is now particularly important, as it is used in the Oxford–AstraZeneca COVID-19 vaccine, which has been administered to >500 million people. We and others used Evans’ A438 liquid formulation for species E vectors [4], but to our knowledge, data on the stability in this formulation have not been published. Similarly, we are unaware of published data on the lyophilisation of these vectors, although there are some data on other approaches to the drying of AdC68 [21,22].

Here, we evaluate the stability of two serotypes of species E adenovirus (ChAdOx1 and ChAdOx2) for up to a year in liquid and lyophilised formulations that had been reported to perform well with other adenoviruses. We report the maintenance of potency measured both in vitro using a robust immunostaining-based infectivity assay, and in vivo immunogenicity in mice.

## 2. Materials and Methods

### 2.1. Viral Preparation, Titration, and Storage

The design and initial production of ChAdOx1 RVF GnGc (ChAdOx1 RVF) and ChAdOx2 RabG were previously described [21,23]. Methods of viral preparation and particle titration for each experiment are summarised in Table 1. Infectivity measurement was performed using a hexon immunostaining assay as previously described [18] except for the following modifications: titrations were performed using HEK293 T-REx cells (Thermo Scientific) rather than HEK293A, and the primary used antibody was clone B025/AD51 (GeneTex).

**Table 1 vaccines-09-01249-t001:** Virus production and titration methods.

Virus and Formulation Type	**Figures**	Viral Production Process	Starting Infectivity Titer (IU/mL)	Starting Viral Particle Titer (VP/mL)	VP Assay Method
ChAdOx1 RVF, liquid	Figure 1A, Figure 2A,B and Figure 3A–D	No chromatography, inhouse	1.5 × 10^9^	2.1 × 10^11^	qPCR, inhouse [24]
ChAdOx2 RabG, liquid	Figure 1B and Figure 2A,B	With chromatography	2.2 × 10^9^	2.2 × 10^11^	Spectrophotometry [25]
ChAdOx1 RVF, liquid	Figure 2C	With chromatography	2.3 × 10^9^	Not performed	Not performed
ChAdOx2 RabG, liquid	Figure 2C	With chromatography	2.20 × 10^9^	2.2 × 10^11^	Spectrophotometry [25]
ChAdOx1 RVF for lyophilisation	Figure 4A,C,D	No chromatography, advent	3.0 × 10^9^	1.0 × 10^11^	Proprietary qPCR
ChAdOx2 RabG for lyophilisation	Figure 4B	With chromatography	1.8 × 10^9^ IU/mL	1.3 × 10^11^	Spectrophotometry [25]

Durations and temperatures of viral storage were as indicated in the descriptions of individual experiments. Humidity during storage was not controlled or monitored, but all storage was in airtight, crimped glass vials.

### 2.2. Viral Production and Titration Methods

Methods used to produce and titre the virus used to produce each figure are summarised in Table 1. Starting titres shown are those after any dilution steps, i.e., immediately prior to the start of a stability study or lyophilisation. The ‘with chromatography’ viral production process was inhouse and as previously described [24]. This process was designed for human clinical purposes and incorporates an anion exchange chromatography step, preceded and followed by two tangential flow filtration steps. With the exception of the virus used to produce the data shown in Figure 2C, ChAdOx1 RVF was produced using a process designed to prepare vaccines for veterinary use: the downstream process included a single tangential flow filtration (TFF) step and no chromatography. Through the use of a high-molecular-weight cut-off membrane (300 kDa), this TFF removes a large proportion of host cell residuals. It also diafilters the product into a storage buffer. When performed inhouse, the virus was diafiltered into an A438 formulation buffer (35 mM NaCl, 10 mM histidine, 1 mM MgCl^2^, 0.1 mM EDTA, 0.1% *w/v* polysorbate 80, 7.5% *w/v* sucrose, 0.5% *v/v* ethanol, pH 6.6) [6]. The ChAdOx1 RVF that was used for lyophilisation was thus produced by Advent Srl (Pomezia, Italy), under GMP conditions. In this case, the storage buffer was 5% sucrose, 200 mM NaCl, 2 mM MgCl_2_, 50 mM Tris, pH 8), and the VP titre was measured using a proprietary validated qPCR assay.

### 2.3. Lyophilisation

Lyophilisation was performed by a contract development and manufacturing organisation (ProJect Pharmaceutics, Germany) using a pilot-scale freeze-dryer (Hof Sonderanlagenbau) equipped with capacitance and Pirani pressure sensors. Residual moisture was measured by coulometric Karl Fischer analysis (756 instrument from Mettler Toledo).

The bulk drug substance was diluted 10% (*v/v*) into an excipient solution comprising 5% inulin (*w/v*), 5% mannitol (*w/v*), 100 mM NaCl, 1 mM MgCl2, 10 mM Tris pH 8.2 (as reported by Chen et al. [15]). The expected titre of the formulated product was 3.6 × 10^9^ IU/mL. Then, 2 mL of the formulated product was filled into heat-sterilised ISO 4R glass vials and partially stoppered, with 400 such vials placed in stainless steel racks and then placed within Lyoprotect bags (Teclen).

Cycle parameters were as shown in Table 2. These parameters were modified from those reported by Chen and colleagues with the extension of the freezing step from 3 to 5 h, shortening the primary drying phase from 60 to 5 h, and the extension of the tertiary drying phase from 8 to 45 h.

ChAdOx2 RabG was similarly formulated and dried, but with a 0.5 mL filling volume in a 2.5 mL capacity vial, and using a Virtis Advantage freeze drier (SP Scientific).

The lyophilised vaccine was reconstituted for infectivity titration or immunisation with phosphate-buffered saline (150 mM, pH 7.4) and vortexed for 2 s.

### 2.4. Animal Studies

All animal work was performed in accordance with the U.K. Animals (Scientific Procedures) Act 1986 (ASPA), and was approved by the University of Oxford Animal Welfare and Ethical Review Body (in its review of the application for the UK Home Office Project Licence P9804B4F1).

Female Balb/c (Envigo) and CD-1 (Charles River and Envigo) were housed in a specific-pathogen-free facility and were 6–8 weeks old at the initiation of each experiment. Mice were vaccinated with 1 × 10^5^, 1 × 10^6^, or 1 × 10^7^ IU of ChAdOx1 RVF stored at each temperature (*n* = 6 per dose and temperature). All vaccinations were diluted in PBS to 50 μL and administered intramuscularly, split equally between the gastrocnemius muscles of each hind limb. Serum samples were obtained by cardiac puncture under terminal anaesthesia 4 weeks, 12 months, or 16 months after vaccination, followed by euthanasia by cervical dislocation.

### 2.5. Enzyme Linked Immunosorbent Assays

Recombinant RVFV Gn ectodomain (UniProt accession number P21401, residues 154–560) and Gc ectodomain (UniProt accession number P21401, residues 691–1120) were expressed in HEK293 cells as previously described [26]. This protein was used to perform ELISAs as previously described [26] with the exception of the following modifications. A positive reference serum made from a pool of high responding mice was included on each plate as a standard curve (titrated 1:2 and added in duplicate). The secondary antibody used (goat antimouse whole IgG-alkaline phosphatase, Sigma, St. Louis, MO, USA) was diluted 1:1000 in casein. Plates were developed by adding 100 µL/well of 4-nitrophenyl phosphate disodium salt Hexahydrate (Sigma) in diethanolamine buffer (ThermoScientific). Optical density (OD) was read at 405 nm using a BioTek ELx808 plate reader until the internal control (reference serum diluted 1:800) had reached an OD of 1. OD values of reference serum titrations were then fitted to a 4-parameter standard curve using Gen5 software (v3.09 BioTek). Test sera antibody units were calculated from their OD values using the estimated parameters from the standard curve.

### 2.6. Statistical Analysis

Analyses were performed using Prism 9.0 software (GraphPad). Details of analyses are contained in the figure legends.

## 3. Results

### 3.1. Stability in Liquid Formulation

ChAdOx1 GnGc is a candidate vaccine against Rift Valley fever (RVF) based upon the ChAdOx1 vector backbone derived from chimpanzee adenovirus serotype Y25 [18,23], delivering a transgene encoding the RVF viral envelope glycoproteins n and c; henceforth, the vaccine is referred to as ChAdOx1 RVF, and the proteins are referred to as Gn and Gc. ChAdOx2 RabG is a candidate vaccine against rabies based upon the ChAdOx2 vector backbone that is derived from chimpanzee adenovirus serotype 68 (AdC68) [19,21], delivering a transgene encoding the rabies viral glycoprotein. Both are currently being evaluated in Phase I clinical trials (NCT04754776, NCT04162600). ChAdOx1 RVF is also being evaluated for veterinary use [27,28].

For this study, as for much of our and others’ previous work with species E vectors, both vaccines were formulated in the A438 buffer (35 mM NaCl, 10 mM histidine, 1 mM MgCl2, 0.1 mM EDTA, 0.1% *w/v* polysorbate 80, 7.5% *w/v* sucrose, 0.5% *v/v* ethanol, pH 6.6) [4,6,24] at clinically relevant concentrations of 1–2.5 × 10^11^ VP/mL (for full details, see Section 2).

We studied the stability of the two vaccines at a temperature range from 4 to 45 °C over a period of a year. First, we measured potency by in vitro infectivity assay (Figure 1). Throughout this study, loss of infectivity is reported as ‘log10fold’, i.e., change in the log10-transformed titre. Results were similar for the two products. There was very slow loss of infectivity at 4 °C, compatible with at least 6 month shelf life at 2–8 °C. After 6 months, infectivity loss was ≤0.1 log10 fold (within the margin of assay variability). After one year, infectivity loss was 0.1 log10 fold for ChAdOx1 RVF, and 0.3 log10 fold for ChAdOx2 RabG.

**Figure 1 vaccines-09-01249-f001:**
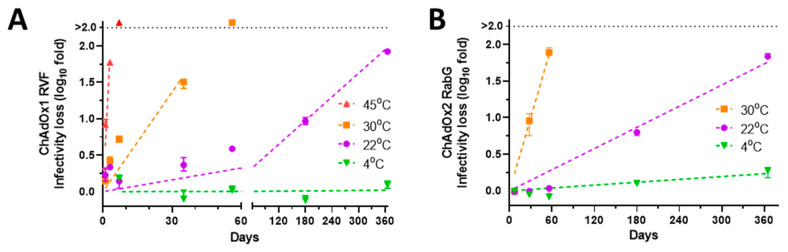
Stability of (**A**) ChAdOx1 RVF (X48A) and (**B**) ChAdOxRabG ( X48D) in liquid formulation. Results presented in terms of infectivity loss from baseline titre. Points, median; error bars, range of triplicate vaccine vials, each titred in duplicate assay wells.

As expected, loss of infectivity was more rapid at higher temperatures: linearity of the Arrhenius plot of the relationship between temperature and rate of infectivity loss suggested inactivation through a reaction with first-order kinetics (Figure 2A,B). Loss of c. 0.15 log10 fold/month at 22 °C supports the suitability of these formulations for short periods (days) outside refrigerators in temperate climates, although rapid loss at 30–45 °C suggests the need for caution in warmer climates.

Sensitivity to freeze–thaw is a concern for some other vaccine products, although adenoviruses are generally regarded as being relatively robust upon freezing. We were unable to detect any loss of infectivity of either product over five freeze–thaw cycles (Figure 2C).

**Figure 2 vaccines-09-01249-f002:**
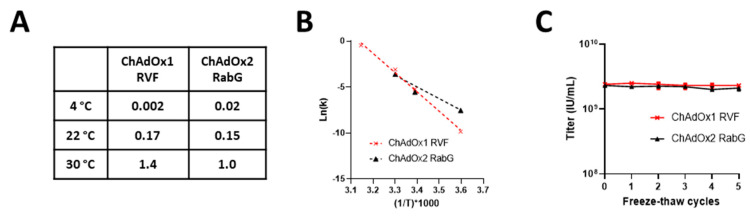
Stability of (**A**) ChAdOx1 RVF (X48A) and (**B**) ChAdOxRabG (X48D) in liquid formulation. (**A**) Tabulated rates of infectivity loss (k = reduction in log10 (titre) per month calculated by linear regression using data shown in Figure 1A,B) for the two viruses at three temperatures; (**B**) the same data in the form of an Arrhenius plot; (**C**) infectivity of samples of the two viruses over a series of freeze–thaw cycles.

In parallel with the in vitro potency measurements, we evaluated the in vivo potency of ChAdOx1 RVF. Mice were immunised intramuscularly with material that had been subject to storage at various temperatures during the stability studies described above, and with comparator material that had been stored at −80 °C. For each storage condition, three doses were used to characterise the dose–response relationship. Serum was collected 28 days later, and antibody titres to RVF Gn and Gc proteins were measured by ELISA (Figure 3A–D). Broadly, the results of in vivo studies were consistent with the findings from in vitro work. Immunogenicity was maintained after storage in A438 at 20 °C for 1 month or 4 °C for 12 months. On the other hand, statistically significant deterioration of immunogenicity was apparent after storage at 30 °C for 1 month or 20 °C for 12 months. Results were similar for Gn and Gc.

**Figure 3 vaccines-09-01249-f003:**
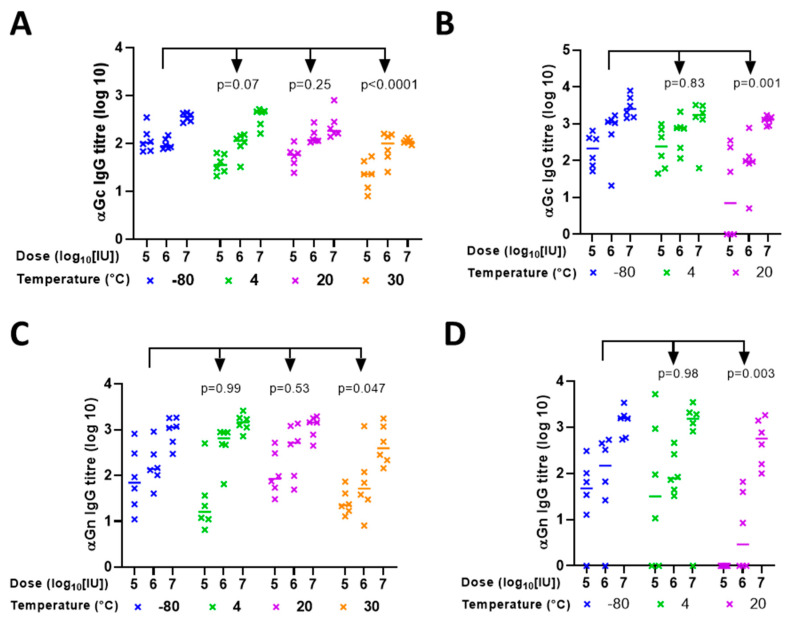
In vivo potency (immunogenicity in mice) of ChAdOx1 RVF after storage for (**A**,**C**) 1 month or (**B**,**D**) 12 months in liquid formulation. X axes indicate dose (based upon the starting titre, i.e., assuming no loss of potency in storage) and storage temperature. Y axes indicate ELISA-measured antibody titres against RVF (**A**,**B**) Gc and (**C**,**D**) Gn by ELISA. Points, individual mice; lines, group medians. To present global analysis (across dose levels) of the effect of temperature upon each readout, *p* values are from 2-way ANOVA of log10-transformed ELISA titres with dose and temperature as factors, and Dunnett’s post-test for multiple comparison of each temperature condition to the −80 °C condition as a control. Significant interaction between the effects of dose and temperature was seen for ANOVA relating to (B) (*p* = 0.004; *p* for interaction was >0.05 for all other ANOVA analyses), and residuals were non-normal for some analyses. In view of this, two alternative analyses were performed, both of which provided similar results. First, data were analysed by Kruskal–Wallis test within dose levels, with similar results (data not shown). Second, 4-parameter dose–response curves were fitted by nonlinear regression to dose–response curves showing the relationship of log10 (titre) to log10 (dose), constraining curves to share a maximal response, zero as the lowest response, and slope, and testing for equality of EC50 values. Results were again similar to the original 2-way ANOVA.

### 3.2. Lyophilisation

To test the stability of our vectors in a lyophilised formulation, we based our work upon a report by Chen et al. [15]. From our review of the peer-reviewed literature, this manuscript provides the most encouraging data on adenoviral lyophilisation, which was accompanied by sufficient methodological information to allow for the reproduction of the formulation and process, and sufficient detail to interpret and have confidence in the obtained data. Chen et al. reported that, using a vector based upon human adenovirus serotype 5 in a formulation comprising 5% inulin (*w/v*), 5% mannitol (*w/v*), 100 mM NaCl, 1 mM MgCl^2^, 10 mM Tris pH 8.2, inprocess infectivity loss of ~0.3log10 could be obtained, with <0.3 log10 of further loss over a month of storage at 37 °C.

Initial work undertaken by our CRO partner (ProJect Pharmaceutics, Germany) sought to shorten Chen’s long (60 h) primary drying step, and incorporated further cycle modifications in order to achieve satisfactory cake appearance. Drying the veterinary formulation of ChAdOx1 RVF using the resulting cycle achieved inprocess infectivity loss 0.4 log10-fold and was judged on this basis to not have affected product quality. We subsequently similarly lyophilised ChAdOx2 RabG.

Figure 4A,B show in vitro infectivity results obtained with the two lyophilised products, while Figure 4C,D show in vivo immunogenicity of the products after 16 months in storage at various temperatures.

**Figure 4 vaccines-09-01249-f004:**
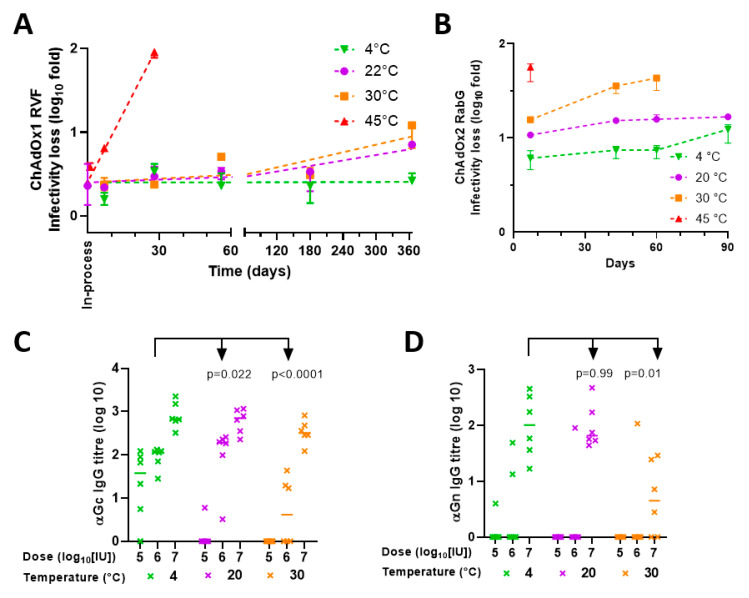
Stability of ChAdOx1 RVF and ChAdOx2 RabG in lyophilised formulation. (**A**) Stability of ChAdOx1 RVF; results presented in terms of infectivity loss from the baseline (prelyophilisation) titre. Points, median; error bars, range of triplicate vaccine vials, each titred in duplicate assay wells. Linear regression fitted lines shown. Inprocess loss was 0.4 log10-fold for ChAdOx1 RVG. (**B**) Stability of ChAdOx2 RabG in lyophilised formulation, presented as (**A**) (without linear regression fits in view of limited data). Inprocess loss estimated at c. 1 log10-fold. (**C**,**D**) In vivo potency of ChAdOx1 RVF after storage for 16 months in lyophilised formulation at indicated temperatures assessed by ELISA measurement of titres against (**C**) Gc and ( **D**) Gn. *p* values are from 2-way ANOVA for (**C**). For Gn data (**D**), statistical analysis was only performed for data from 1 × 10^7^ IU groups (in view of the large number of zero values at lower doses, even with 4 °C-stored virus): *p* values in (**D**) are from a Kruskal–Wallis test with Dunn’s multiple-comparison test versus the −80 °C groups.

Inprocess losses were 0.4 log10-fold for ChAdOx1 RVF and 0.8 log10-fold for ChAdOx2 RabG. For both products, there was <0.5 log10-fold infectivity loss in storage at 20 °C over one year, although total loss (combining instorage and inprocess loss) was higher (around or exceeding 1 log10). There was relatively little change in the immunogenicity of ChAdOx1 RVF stored at 22 °C as compared to storage at 4 °C. Although there was a decrease in marginal statistical significance in Gc ELISA driven by decreased seroconversion at the lowest dose (Figure 4C), no effect was seen on Gn ELISA (Figure 4D).

Overall, studies investigating stability in storage at 30 °C suggested that the lyophilised formulation was not suitable to achieve long-term storage under these conditions. Although the infectivity assay did not detect substantially accelerated loss of potency of ChAdOx1 RVF at 30 °C as compared to 20 °C (Figure 4A), loss of potency was clearly apparent when immunogenicity was tested after the product had been stored at 30 °C for 16 months (Figure 4C,D). This was in contrast to the marginal effect upon immunogenicity of storage at 20 °C, as above. Relatively rapid loss of in vitro potency of ChAdOx2 RabG at 30 °C was also apparent (Figure 4B). Loss of titre at 45 °C was rapid for both ChAdOx1 RVF and ChAdOx2 RabG (Figure 4A,B). Table 3 summarises the animal experiments and in vivo immunogenicity data.

## 4. Discussion

The results we report here with ChAdOx1 and ChAdOx2 in liquid formulations are reassuringly similar to those previously reported with other serotypes in similar buffers. Our data thus support the distribution and storage of products based upon species E vectors in such formulations at 2–8 °C. This is current practice for commercially supplied ChAdOx1 nCoV-19 (AstraZeneca/Serum Institute of India). Given that stability is largely independent of the encoded transgene, it is reasonable for investigators to assume that other candidates in development using the same platform would be similarly stable.

Using a previously reported formulation for lyophilisation, with some modifications of the drying cycle parameters, we demonstrated enhanced stability of the vectors in storage at 22–30 °C (as compared to liquid formulations). With further characterisation of the stability profile, this formulation may be suitable for storage for a number of months at 22 °C, though possibly not for a full year. During the drying process, however, potency loss of ≥0.4 log10-fold was noted. The acceptability of such inprocess loss would depend upon whether it could be shown to be consistent between batches (resulting in a consistent potency of the released product), and the economics of drug substance manufacture (i.e., the cost of overfilling in anticipation of lost potency). We did not directly test the effect of our drying cycle modifications upon instorage stability (as distinct from inprocess loss). The use of alternative cycle parameters (including those previously reported [15]) may improve the performance of this formulation. More broadly, there remains significant scope for the optimisation of adenoviral lyophilisation to reduce inprocess loss, improve stability in storage at higher temperatures, and develop shorter and more economical lyophilisation cycles.

In vivo immunogenicity essentially confirmed our in vitro stability data. Large numbers of animals are required to establish dose–response curves and hence properly quantify changes in in vivo potency; even using large numbers, it remains challenging to achieve the level of precision offered by in vitro assays. On ethical and practical grounds, we therefore suggest that in vitro infectivity is used as the primary potency assay for adenovirus vectored vaccines in future, coupled to properly understanding the dose–response relationship in the target species.

Clinical and veterinary trials of adenovirus vectored vaccines have not established a wide therapeutic window (i.e., range of doses over which efficacy can be achieved, with acceptable reactogenicity). Doses typically used to demonstrate efficacy may be at the upper limit of the tolerable range. For example, with ChAdOx1 nCoV-19, reactogenicity is appreciably higher at the licensed dose of 5 × 10^10^ VP than at 2.2 × 10^10^ VP, and the product specification for dose as determined by VP titre is 3.5–6.5 × 10^10^ VP [2]. The parameter used to define the dose (VP) is linked to the main characteristic indicating storage stability (IU titre) by the P:I ratio, and the specification for P:I ratio must be similarly tight. There is thus a limited window between the maximal safe release potency and the acceptable lower limit of potency at the end of storage, and hence more limited scope to allow for instorage losses by overfilling than is the case for some other vaccines. The difference between the minimal efficacious release potency and the minimal acceptable potency at the end of storage (i.e., the acceptable instorage loss of potency) is even tighter. The maximal acceptable potency loss for regulatory approval of a storage condition with such vaccines is often <0.5 log10-fold. Greater instorage loss is only likely to be acceptable if clinical data can be provided demonstrating acceptable immunogenicity (or ideally efficacy) at a more substantially lower dose than the minimal release potency.

On the basis of this cut-off of c. 0.5 log10-fold acceptable potency loss, our data suggest that both the liquid and lyophilised formulations tested here could enable ‘last leg’ distribution under Extended Controlled Temperature Chain conditions as defined by the WHO [29]. This term refers to definition, on an individual product basis, of permissible temperature ranges and short-term periods (usually a few days) for which an individual product may be kept outside the conventional 2–8 °C refrigerated ‘cold chain’ immediately prior to administration. The calculated rates of potency loss of liquid formulations of both vaccines would suggest suitability for storage at 20 °C for a period of no more than 3 days, dependent upon the acceptable level of loss for a particular product. The stability of the lyophilised formulations, on the other hand, might enable periods of 3 days or potentially longer at 30 °C. Suitability for distribution under these conditions could be of substantial benefit in resource-limited settings.

In this study, we characterised the infectivity and subsequent immunogenicity in mice of two species E simian adenovirus-vectored vaccines when stored at a range of temperatures. Both ChAdOx1 RVF and ChAdOx2 RabG are currently in clinical trials, and our data support the use of liquid formulations of these vectors for storage at 2–8 °C for up to 1 year, with potential for short-term storage at higher temperatures. Given that the thermostability of these vaccines is largely independent of the encoded transgene, similar results may be obtained with other candidate vaccines using the chimpanzee adenovirus vaccine platform.

## Figures and Tables

**Table 2 vaccines-09-01249-t002:** Lyophilisation cycle parameters.

Step		Shelf Temperature	Ice Condenser Temperature	Pressure (Pirani)	Time Step	Cumulative Time
**#**	**Description**	**(°C)**	**(°C)**	**(mbar)**	**(h:min)**	**(h:min)**
**1**	Loading	5	-	atm	00:01	0:01
**2**	Freezing	−50	-	atm	00:55	0:56
**3**	Freezing	−50	-	atm	05:00	5:56
**4**	Vacuum adjustment	−50	−70	0.03	00:30	6:26
**5**	Primary drying	−50	−70	0.03	05:00	11:27
**6**	Secondary drying ramp	0	−70	0.03	00:50	12:17
**7**	Secondary drying	0	−70	0.03	04:00	16:17
**8**	Tertiary drying ramp	20	−70	0.03	00:20	16:37
**9**	Tertiary drying	20	−70	750	45:00	61:37
**10**	Venting (N_2_)	20	−70	atm	00:05	61:42

**Table 3 vaccines-09-01249-t003:** Summary of ChAdOx1 RVF in vivo immunogenicity.

Vaccine	ELISA Antigen	Time since Vaccination	*n* ^2^	Dose (IU)	Immunogenicity in Mice (Log Change) ^1^						
					Liquid				Lyophilised		
					−80 °C	4 °C	20 °C	30 °C	4 °C	20 °C	30 °C
ChAdOx1 RVF	Gc	1 month	24	1 × 10^5^	0.0	−0.5	−0.3	−0.7	*-*	*-*	*-*
			24	1 × 10^6^	0.0	0.1	0.1	0.1	*-*	*-*	*-*
			24	1 × 10^7^	0.0	0.1	−0.3	−0.5	*-*	*-*	*-*
		12 months	18	1 × 10^5^	0.0	0.1	−1.5	*-*	*-*	*-*	*-*
			18	1 × 10^6^	0.0	−0.2	−1.1	*-*	*-*	*-*	*-*
			18	1 × 10^7^	0.0	−0.2	−0.3	*-*	*-*	*-*	*-*
		16 months	18	1 × 10^5^	*-*	*-*	*-*	*-*	0.0	−1.6	−1.6
			18	1 × 10^6^	*-*	*-*	*-*	*-*	0.0	0.2	−1.4
			18	1 × 10^7^	*-*	*-*	*-*	*-*	0.0	0.0	−0.3
	Gn	1 month	24	1 × 10^5^	0.0	−0.6	0.1	−0.5	*-*	*-*	*-*
			24	1 × 10^6^	0.0	0.7	0.6	−0.4	*-*	*-*	*-*
			24	1 × 10^7^	0.0	0.1	0.1	−0.5	*-*	*-*	*-*
		12 months	18	1 × 10^5^	0.0	−0.2	−1.7	*-*	*-*	*-*	*-*
			18	1 × 10^6^	0.0	−0.3	−1.7	*-*	*-*	*-*	*-*
			18	1 × 10^7^	0.0	0.0	−0.4	*-*	*-*	*-*	*-*
		16 months	18	1 × 10^5^	*-*	*-*	-	-	U ^3^	U	U
			18	1 × 10^6^	*-*	*-*	*-*	*-*	U	U	U
			18	1 × 10^7^	*-*	*-*	*-*	*-*	0.0	−0.2	−1.4

^1^ Median log change in antibody titre from six mice in each group. Log change for liquid formulations is in relation to vaccine stored at −80 °C. Log change for lyophilised formulations is in relation to vaccine stored at 4 °C. ^2^ number of mice in total per row (for each temperature, *n* = 6). Gn and Gc ELISA responses were measured in the same mice, so they do not represent additional animals. ^3^ U = undetectable antibody response.

## Data Availability

The authors declare that the data supporting the findings of this study are available within the paper.

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
