# Peer review of "Stability of Chimpanzee Adenovirus Vectored Vaccines (ChAdOx1 and ChAdOx2) in Liquid and Lyophilised Formulations"

_vaccines, 2021, doi:10.3390/vaccines9111249_

Round 1
Reviewer 1 Report
It was a pleasure to read the research article on “Stability of chimpanzee adenovirus vectored vaccines in liquid and lyophilized formulations”. This is a good research study on the stability evaluation of two chimpanzee adenoviruses ChAdOx1 and ChAdOx2 in solution form and after lyophilization. The article produces some valuable date that are useful for the vaccine’s stability. The article is well written in terms of grammar and English; however, the organization of the article need to be changed. I would suggest the following changes of the article to be published.
- The title of article should be more specific. I would suggest “Evaluation of stability of chimpanzee adenovirus (ChAdOx1 and ChAdOx2) vectored vaccines in liquid and lyophilized formulations”.
- Figures are placed all together in one page which is not easy to follow or read the captions for the readers. Therefore, it would be better if the figures are separated in terms of individual groups. Figure can be divided in group of in vivo study, lyophilization stability etc.
- For the stability test, it’s not easy to understand how many samples are there and what are the different conditions. A table indicating the different groups in stability test would give a clear picture. The table can contain temperature, time, percent change of immunogenicity.
- The general condition of stability test includes temperature and humidity. The article mentioned the temperature; however, it should also mention the humidity. should mention the percentage of humidity in stability condition which can cause the hydrolysis and oxidation.
- The animal study is not clearly described, how many groups were there, is there any control group, was the animal study conducted just one time or at different time point such as one month and twelve months. These can be placed in a table.
- In line 331-333, please specify the number of days instead of writing ‘few days’.
- Change the sentence in line 287-288.
- The article does have any conclusion summarizing the whole work. It would be better to have a conclusion of the article.
Author Response
We thank the reviewer very much for their comments and have addressed the individual points below.
- The title of article should be more specific. I would suggest “Evaluation of stability of chimpanzee adenovirus (ChAdOx1 and ChAdOx2) vectored vaccines in liquid and lyophilized formulations”.
We have amended the title as suggested.
- Figures are placed all together in one page which is not easy to follow or read the captions for the readers. Therefore, it would be better if the figures are separated in terms of individual groups. Figure can be divided in group of in vivo study, lyophilization stability etc.
We have split figure 1 into three separate figures. There are now no more than four sub-panels in any of the figures. We hope the reviewer will agree the new structure is clear and logical.
- For the stability test, it’s not easy to understand how many samples are there and what are the different conditions. A table indicating the different groups in stability test would give a clear picture. The table can contain temperature, time, percent change of immunogenicity.
We have added a new table (Table 3) which summarises the animal experiments and provides the requested information.
- The general condition of stability test includes temperature and humidity. The article mentioned the temperature; however, it should also mention the humidity. should mention the percentage of humidity in stability condition which can cause the hydrolysis and oxidation.
We have added explanation of the storage conditions (lines 98-100), stating that humidity during storage was not controlled or monitored, but all storage was in airtight, crimped glass vials (and hence environmental humidity was not relevant).
- The animal study is not clearly described, how many groups were there, is there any control group, was the animal study conducted just one time or at different time point such as one month and twelve months. These can be placed in a table.
We have clarified doses and animal numbers in methods. We hope that the addition of table 3 in response to point 4 will also help to make these details clear.
- In line 331-333, please specify the number of days instead of writing ‘few days’.
We have re-worded the relevant section (now at lines 358-360 in the revised manuscript) although we do not feel it is meaningful to be categorical about this, given that the acceptable level of loss would vary dependent upon the release potency & acceptable minimum potency of a specific product.
- Change the sentence in line 287-288.
We have reworded this for clarity and hope the revised version is acceptable.
- The article does have any conclusion summarizing the whole work. It would be better to have a conclusion of the article.
We have now added a concluding paragraph.
Reviewer 2 Report
The authors studied the stability of two candidate vaccines for two species E serotypes: a Rift Valley fever vaccine based upon the ChAdOx1 vector (Y25 serotype) used in ChAdOx1 nCoV-19, and a rabies vaccine based upon a ChAdOx2 vector (AdC68 serotype). The in vitro and in vivo potency measurements were made to test the stability in liquid and lyophilized formulations. Their findings claim that supports the suitability of liquid formulations of these vectors for storage at 2-8 °C for up to 1 year, and potentially for non-refrigerated storage for a brief period during last-leg distribution (perhaps 1-3 days at 20 °C – precise definition of acceptable last-leg storage conditions would require further product-specific data). The article can be published after the revision of the following scientific points:
- The authors should fetch some latest references in their problem statement especially among 4- 21.
- In Figure 1, the C, D and E can be given as separate figure with more clarity.
Overall article is written well still its English language can be improved by proofreading and checking for typos
Author Response
We thank the reviewer very much for their comments and have addressed the individual points below.
- The authors should fetch some latest references in their problem statement especially among 4- 21.
To our knowledge, the references cited in the introduction (including 4-21) remain the most appropriate sources relevant to the area, spanning the years 2001-2020. We would be pleased to consider citing any specific sources which the reviewer and editor feel are important omissions.
- In Figure 1, the C, D and E can be given as separate figure with more clarity.
See response to reviewer #1, comment 2.
- Overall article is written well still its English language can be improved by proofreading and checking for typos"
We have made a number of minor corrections throughout the text (with tracked changes). We cannot find any further errors but would of course be pleased to correct any further errors brought to our attention.